# Personalised Tasted Masked Chewable 3D Printed Fruit-Chews for Paediatric Patients

**DOI:** 10.3390/pharmaceutics13081301

**Published:** 2021-08-20

**Authors:** Atabak Ghanizadeh Tabriz, Daniel Henri George Fullbrook, Lilian Vilain, Youri Derrar, Uttom Nandi, Clara Grau, Anaïs Morales, Gemma Hooper, Zoltan Hiezl, Dennis Douroumis

**Affiliations:** 1Faculty of Engineering and Science, School of Science, University of Greenwich, Chatham Maritime, Chatham, Kent ME4 4TB, UK; ata_ghanizadeh@hotmail.com (A.G.T.); df1312s@greenwich.ac.uk (D.H.G.F.); g.hooper@gre.ac.uk (G.H.); z.hiezl@gre.ac.uk (Z.H.); 2Polytech Marseille, School of Engineering, Aix Marseille Université, 163 Avenue of Luminy, 13009 Marseille, France; lilian.vilain@etu.univ-amu.fr (L.V.); youri.derrar@etu.univ-amu.fr (Y.D.); 3Medway School of Pharmacy, University of Kent, Chatham Maritime, Chatham, Kent ME4 4TB, UK; u.nandi@kent.ac.uk; 4School of Chemistry of Mulhouse (ENSCMu), University of Haute-Alsace (UHA), 3 Street Alfred Werner, 68093 Mulhouse, France; clara.grau@uha.fr (C.G.); anais.morales@uha.fr (A.M.)

**Keywords:** personalised dosage forms, 3D printing, paediatric, flavours, sweeteners, taste masking, palatability, sensory evaluation

## Abstract

The development of personalised paediatric dosage forms using 3D printing technologies has gained significant interest over the last few years. In the current study extruded filaments of the highly bitter Diphenhydramine Hydrochloride (DPH) were fabricated by using suitable hydrophilic carries such as hydroxypropyl cellulose (Klucel ELF^TM^) and a non-ionic surfactant (Gelucire 48/16^TM^) combined with sweetener (Sucralose) and strawberry flavour grades. The thermoplastic filaments were used to print 3D fruit-chew designs by Fused Deposition Modelling (FDM) technology. Physicochemical characterisation confirmed the formation of glass solution where DPH was molecularly dispersed within the hydrophilic carriers. DPH was released rapidly from the 3D printed fruit-chew designs with >85% within the first 30 min. Trained panellists performed a full taste and sensory evaluation of the sweetener intensity and the strawberry aroma. The evaluation showed complete taste masking of the bitter DPH and revealed a synergistic effect of the sweetener and the strawberry flavour with enhanced sweet strawberry, fruity and aftertaste perception. The findings of the study can be used for the development of paediatric dosage forms with enhanced organoleptic properties, palatability and medication adherence.

## 1. Introduction

Paediatric patients are a special population that requires specific considerations when designing drug products due to the need for clinical safety and medication adherence [1]. Very often the lack of therapy adherence leads to deprived health consequences for paediatric patients. In addition, most pharmaceutical products are not indicated for paediatric applications as clinical trials in the pre-development stage exclude children. Thus, the European Union recognises the need for paediatric-centric medicines and introduced the paediatric investigation plan and several guidelines for pharmaceutical development [2] and clinical trials for paediatric products [3,4].

The administration of oral dosage forms is the most acceptable for school children but also for infants (mini-tablets < 2 mm), especially those that can be easily swallowed and disintegrate in the oral cavity. Organoleptic properties such as appearance, taste, texture or smell are also important including and promote adherence in paediatric populations. Another consideration in the development of paediatric products is the selection of the excipients which should be approved for paediatric patients and thus the formulations should be carefully designed [5].

Overall, the development of such pharmaceutical products with high quality, safety and clinical efficacy is extremely challenging and requires the implementation of novel manufacturing technologies which can address the aforementioned challenges but also to produce personalised dosage forms that fit the patient’s clinical needs (e.g., dose, pharmacokinetics, palatability).

A disruptive technology that has attracted enormous interest in pharmaceutical research over the last few years is 3D printing or also known as additive manufacturing. It has especially found ground for paediatric dosage forms due to its flexibility to fabricate precise dosage forms with complex geometries but also with acceptable palatability for children [6,7]. Scoutaris et al. were the first to introduce 3D printed paediatric formulations with excellent taste masking, precise drug doses, immediate release and complex geometries by imitating Starmix^®^ sweets and print ”candy-like” designs [8]. Nevertheless, 3D printing has been studied systematically for the optimisation of 3D printed tablets [9] using a variety of technologies and materials (e.g., polymers, resins, hydrogels). The selection of the technology is usually related to the drug product specifications and such as water solubility, thermal stability, active dose and drug-excipient miscibility [10,11]. Selective laser sintering (SLS) was used by Awad et al. to develop fast dissolving tablets of paracetamol for visually impaired patients. The tablets presented fast disintegrating times while it was feasible to design braille and moon patterns on the tablet surface [12]. The same group printed orodispersible tablets loaded with ondansetron-cyclodextrin complexes using SLS printing and compared the results with the marketed product Vonau^®^ Flash.

Stereolithography (SLA) is also an attractive 3D printing technology that has been implemented for the development of oral solid dosages by using suitable photocurable resins of poly (ethylene glycol) diacrylate-poly (caprolactone) triol at various ratios [13]. The printed structures showed sustained release of paracetamol and aspiring for 24 h. The technology was recently employed for the printing of multilayer tablets (polypills) by incorporating 6 active ingredients naproxen, prednisolone, chloramphenicol, caffeine, paracetamol and aspirin [14]. The technology allowed the fabrication of tablets with different geometries and variable drug loadings.

Fused Deposition Modelling (FDM) is another 3D printing technology that has been used extensively due to the flexibility of using a wide range of pharmaceutical-grade polymers [15,16,17]. FDM is coupled with Hot Melt Extrusion (HME) to produce printable filaments loaded with drug substances at various ratios. Among the significant advantages of FDM printed tablets is not only the selection of polymers with various properties (e.g., pH-dependent, non-ionic) but also the possibility to adjust the infill density of the various geometries and hence the dissolution rates [18].

In the current study, HME processing was coupled with FMD for the printing of personalised paediatric fruit-chews with improved organoleptic properties and palatability. In addition, the actual dose can be adjusted according to the patient needs but also to meet the required dissolution profiles and hence pharmacokinetic properties. HME facilitated the successful taste masking of DPH while co-processing of sweetener-strawberry flavour, for the first time, enhanced the sensory and taste perception of the printed constructs. The fruit-chew printed structures were designed to imitate confectionaries by adding colouring.

## 2. Materials and Methods

### 2.1. Materials

Diphenhydramine hydrochloride (DPH, purity > 98%) was purchased from Sigma-Aldrich (Rochester, UK). Strawberry flavouring was purchased from Symrise Ltd. (Marlow, UK), sucralose was purchased from Merck Life Science UK Limited (Dorset, UK). Various food grade (yellow, green, blue and red) colourings were purchased from Ash Spice company (Leicester, UK). Gelucire 48/16^TM^ (GLC) and hydroxypropyl cellulose (Klucel ELF^TM^) were kindly donated by GATTEFOSSE (Lyon, France) and Ashland (Rotterdam, The Netherlands), respectively.

### 2.2. Hot Melt Extrusion for Fabrication of 3D Printing Filaments

The physical mixture containing Klucel ELF^TM^, Gelucire 48/16^TM^, DPH, food colouring, flavouring and sucralose was prepared at a ratio of 80/14.5/2.5/1/1.1/0.9 (wt%) and blended using a turbula shaker-mixer (Glen Mills T2F Shaker/Mixer) at 75 rpm for 10 min to assure the blend is homogenous. The blend was fed into a 16 mm twin-screw extruder (Eurolab 16, Thermo Fisher, Karlsruhe, Germany) at a rate of 300 g/h. The heating zones across the extruder barrel starting from the feeding zone were set to 50 °C, 80 °C, 100 °C, 120 °C, 150 °C, 150 °C, 150 °C and the die temperature was set to 120 °C. The die diameter was set to 3 mm where the extruded filament thickness was maintained within 2.5 to 2.9 by adjusting the conveying belt speed.

### 2.3. Design and 3D Printing of Paediatric Tablet Designs

Paediatric tablet designs (smurf, palm tree, cherry, banana) and normal tablets were created via CATIA v5 software and converted into stl files and sliced by Cura software V 4.0 (Ultimaker, Utrecht, The Netherlands) to generate 3D printing patterns for the printer. Ultimaker 3 extended (Ultimaker, Utrecht, The Netherlands) a dual FDM printer was used as the printing platform to produce the designed structures.

Tablet, palm, cherry and banana designs were printed using a 0.4 mm nozzle at 165 °C. For the smurf structure, the design was divided into two compartments (body and hat) and the first print core was fed with a red colour filament while the second print core with a blue coloured filament, respectively. Both compartments of the smurf were printed using a 0.4 mm nozzle at 165 °C. Similarly, the tablet, palm, cherry and banana designs were printed using a respective coloured filament with 0.4 mm nozzle at 165 °C.

All designs were printed with one covering layer and a 30% infill density. The print speed and print layer height were set to 10 mm/s and 0.1 mm respectively. The build plate temperature was maintained at 65 °C to ensure the adhesion of the tablets across the print time. All printed designs were adjusted to weigh approximately 500 mg where the DPH loading was 12.5 mg.

### 2.4. Thermal Gravimetric Analysis (TGA)

The thermal stability of the bulk materials and the extruded filaments were analysed using a Thermal gravimetric analysis instrument (TGA Q5000 Thermal instruments). Approximately 2–2.5 mg of the samples were weighed and placed into an aluminium pan. The samples were heated from 25 °C to 400 °C at a rate of 10 °C/min. The extracted data were analysed using the TA Universal Analysis software.

### 2.5. Differential Scanning Calorimetry (DSC)

DSC thermograms of the bulk materials and the 3D printing filaments were obtained using a differential scanning calorimeter (Mettler-Toledo 823e, Greifensee, Switzerland). The samples were weighed between 2–2.5 mg in a 40 uL aluminium pan and crimped promptly. The samples were tested from 20 °C to 300 °C at a 10 °C/min heating rate. The extracted data were analysed using STARe Excellence Thermal Analysis Software.

### 2.6. Scanning Electron Microscopy (SEM)

Scanning electron microscopy (Hitachi SU8030, Tokyo, Japan) was used to evaluate the extruded filament’s morphology as well as layer consistency of the 3D printed designs. SEM images of the filaments and the tablets were captured by electron beam accelerating voltage of 2 KV and magnification of 30× and 70×.

### 2.7. X-ray Powder Diffraction (XRPD)

X-ray powder diffraction was utilised to investigate the physical state of the bulk materials and the filaments. The XRPD data were collected using a D8 Advance X-ray Diffractometer (Bruker, Karlsruhe, Germany) equipped with a LynxEye silicon strip position sensitive detector and parallel beam optics. The diffractometer was operated with a transmission geometry using Cu Kα radiation at 40 kV and 40 mA. The instrument was computer-controlled using XRD commander software (Version 2.6.1, Bruker AXS, Karlsruhe, Germany) and the data were analysed using the EVA software (version 5.2.0.3, Bruker AXS, Germany). Samples were placed between foils of 2.5 µm thick mylar for measurement. Data were collected between 5–60° 2θ with a step size of 0.04° and a counting time of 0.2 s per step.

### 2.8. Tensile Testing

A texture analyser TA HD plus (Stable Micro Systems, Surrey, UK) was used to investigate the mechanical properties of the placebo and the drug loaded filaments. Five specimens were tested for each filament formulation. The gauge lengths and the total lengths were set to 8 cm and 12 cm, respectively. The gripping distance at the top and bottom remained at 2 cm. The tensile test was performed at 1 mm/s according to ISO 527 standards. Each filament specimen diameter was measured across 5 locations within the samples and the average diameter was used for data analysis. The specimens were inserted into a 3D printed thermoplastic polyurethane holder and tightened gently with grippers to ensure the stability of the filaments during the tensile test.

### 2.9. In Vitro Dissolution Studies and HPLC Analysis

DPH release from the 3D printed fruit-chews was estimated using a Varian 705 DS (Winston-Salem, NC, USA) paddle apparatus at 50 rpm and 37 ± 1 °C in purified water for 2 hr according to USP guidelines. Samples (5 mL) were collected in triplicates from the dissolution vessels at 10, 20, 30, 45, 60, 90 and 120 min time intervals. The collected samples were filtered by 0.45 μm filters and analysed using high-performance liquid chromatography using Agilent Technologies, 1200 series (Cheshire, UK). An Agilent 1200 series HPLC system consisting of a quaternary gradient pump, autosampler, column oven and a single wavelength detector (UV-vis) was used to analyse DPH. The detector wavelength was set at 258 nm. The chromatographic separations were performed at 30 °C temperature on a Supelco Ascentis Express RP-Amine column ((100 × 4.5 mm), 1.7 μm). The mobile phase was a mixture of 20 mM KH_2_PO_4_ in water: Acetonitrile (70:30, *v/v*), filtered and flowing at the rate of 1 mL/min. The data was collected and analysed with Agilent ChemStation software. An Agilent-Cary 3500 UV-Vis Spectrophotometer with 10 mm matched quartz cells was used to obtain the lambda max of the DPH.

Stock solution (1.0 mg/mL) of DPH was prepared by dissolving 50 mg PRG in 50 mL double-deionised water using a 50 mL volumetric flask. Then calibration standard solutions of 2, 4, 6, 8 and 10 µg/mL of DPH were prepared by diluting the stock solutions with the double-deionised water using 25 mL volumetric flasks.

### 2.10. Taste Masking

Ten healthy panellist volunteers who had given their informed consent first (Ethics Committee of the University of Greenwich, protocol code Ref No: UG09/10.5.5.12, May 2021) were selected and trained for the taste masking, sensory evaluation. For the taste masking evaluation, the panellists held the bulk DPH (the equivalent of 12.5 mg) or the printed fruit-chews for 2 min and then spat out followed by mouth rinsing with H_2_O without swallowing the samples. The bitterness intensity was recorded according to 1–5 scale where 1, 2, 3, 4 and 5 indicate “no”, “threshold”, “slight”, “moderate” and “strong” bitterness.

The sensory training included a full description of the approach, proper use of the scale and confirmation that the samples prepared had sweetness levels that were perceived as “no sweet.”, “sweet”, “very sweet” and “extremely sweet” compared to the reference sample (equivalent sample of sucrose). Similarly, the strawberry aroma was assessed in terms of “sweet”, ”strawberry”, “sour”, “fruity” and “aftertaste” attributes compared to a natural stripped strawberry juice. The panellist studies were conducted according to the World Medical Association’s Code of Ethics (Declaration of Helsinki).

## 3. Results and Discussion

### 3.1. Coupling of Hot Melt Extrusion (HME) and 3D Printing

HME is an established processing technology that has been effectively employed in the optimisation of paediatric dosage forms for the past three decades to increase the solubility of water-insoluble drug substances and taste masking of bitter APIs by forming solid dispersions [19]. HME was used as the primary method for producing printing filaments in this investigation, which were fed into a 3D line for collection in a cartridge and subsequently fed into an FDM 3D printer. HME is the most common method for creating drug loaded filaments [15,20,21] that can then be placed into an FDM 3D printer.

The study’s major goal was to create taste-masked extruded formulations of extruded filaments with improved drug palatability that could be printable in the form of paediatric chewable designs inspired by Starmix (HARIBO plc, Castleford, UK.) confectionaries [8]. The ability of children to swallow oral dosage forms causes great concerns in terms of medication adherence according to Lopez et al. Hence chewable tablets imitating flavour gummy sweets are expected to improve paediatric patient adherence and dose palatability [22]. According to various studies in paediatric populations, chewable tablets are preferred for children from 5–12 years but they are also attractive for adolescents from 10–19 years old [23].

DPH was the chosen model API as it is a well-known paediatric drug substance while it is extremely bitter with unpleasant taste perception [24]. Klucel ELF^TM^ is a thermoplastic material for extruding drug-loaded filaments and presents suitable mechanical properties with suitable printability [25]. In addition, it has been used for immediate release formulations mainly by forming solid dispersions [26]. Gelucire (GLC) grades are made up of mono, di and triglycerides, as well as PEG esters of fatty acids [27,28]. These present a variety of properties depending on their hydrophilic–lipophilic balance and melting point range from 33 to 65 °C. They are used in a wide range of topical and oral and formulations. Solubility enhancement and bioavailability, sustained release, taste masking, protection of APIs from oxygen, light and humidity are all examples of the developed oral formulation [29,30]. Gelucires that comprise of only PEG esters are commonly used in the development of immediate-release dosage forms. Furthermore, the extruded formulation was co-processed by adding sweetener sucralose and strawberry flavour to enhance taste perception and thus palatability.

Sucralose is an artificial disaccharide sweetener that is produced through selective chlorination of sucrose. The combined effect of sweeteners and flavours is well known in the pharmaceutical industry but also in other areas. Several studies have demonstrated the synergy between sweeteners (e.g., ternary mixtures) for sweet taste but also the sweetener/flavour blends for enhanced taste perception [31,32]. According to our knowledge, this is the first time where 3D printed filaments have been fabricated by incorporating sweetener/flavour blends for the development of 3D printed paediatric dosage forms.

For the drug-loaded filaments, DPH, HPC-ELF, GLC, strawberry flavour, sweetener and colouring powder blends were fed and processed into the extruder. The feed rate, screw speed and temperature profile were optimised (data not shown) to fabricate high-quality filaments in terms of surface defects and thickness including mechanical properties (ultimate tensile strength). The HPC-ELF/GLC weight ratio was found to be the critical formulation attribute for the fabrication of printable filaments as GLC was used as a wetting agent. By using various polymer/wetting agent ratios it was found that 15.5% GLC provided the finest filament quality. Various fruit-chews were produced in a variety of designs, as illustrated in Figure 1. To print doses of 12.5 mg DPH per design, Cura software was used to programme the printing with a 30% infill density and printing speed of 10 mm s^−1^. It was found that during the co-printing of the smurf design the non-operating nozzle temperature must be kept at 100 °C to avoid filament degradation due to prolong temperature retention at 165 °C (default settings).

As shown in Table 1 the printing of fruit-chew structures was highly accurate, repeatable and resulted in smooth, defect-free constructions. Despite this, due to minor filament thickness anomalies, weight and content uniformity variation of 2–6% was detected.

Prior to HME processing, the thermal stability of DPH and the excipients was investigated to determine the required extrusion temperatures and hence the barrel temperature profile. As shown in Figure 2 TGA measurements revealed that DPH remained thermally stable below 154 °C while GLC and HPC-ELF at temperatures below 220 °C, after an initial water loss. Similarly, strawberry flavouring and sucralose were found to be thermally stable at 189 °C and 123 °C. However, the extruded filaments after the initial water loss, due to Strawberry flavouring and HPC-ELF, remained thermally stable up to 220 °C. Based on the above results the filament fabrication was conducted at extrusion temperature lower than 150 °C to avoid excipient degradation. Similarly, the printing temperature was set at 165 °C and a build platform temperature of 65 °C for better print adhesion.

### 3.2. Solid State Analysis

To determine the physical form of DPH in the extruded filaments, the X-ray powder diffraction pattern of the filament and each of its separate components were collected (Figure 3). The XRPD diffractogram of HPC-ELF presented broad bumps due to its amorphous nature and GLC two intensity peaks at 19.1° and 23.4/2θ°. The diffractogram of bulk crystalline DPH presented sharp intensity peaks at 10.3°, 12.1°, 18.6°, 20.8°, 21.9/2θ°. For the XRPD of the processed filament, no intensity peaks of DPH could be observed indicating that the active was transformed to its amorphous state. Only two minor peaks at 31.2° and 45.4/2θ°, related to food colouring could be identified (Appendix A). Similarly, the GLC peaks could not be detected on the diffractogram suggesting that it transformed completely to an amorphous state.

As shown in Figure 4 further DSC analysis showed melting endotherms at 169.98 °C and 48.43 °C for DPH and CLC, respectively, while HPC-ELF presented a Tg at 134.90 °C. The filament thermogram showed a single Tg shifted at 128.81 °C and the absence of any DPH melting endotherm indicating that the drug was molecularly dispersed in the lipid/polymer matrix. Previous studies have shown that the presence of glass solutions is accompanied by H-bonding interactions between the drug and the polymer carrier and thus leading to effective taste masking [11,33,34]. The sucralose and food colouring showed melting endotherms at 134.39 °C and 121.82 °C, respectively (Appendix A).

### 3.3. Scanning Electron Microscopy (SEM)

The extrusion processing parameters were tuned to produce filaments with a thickness of 2.5–2.9 mm, allowing for consistent layer-by-layer deposition at an average layer thickness of 100 μm. The width uniformity of the produced filaments was essential for ensuring high-quality prints and preventing under or over extrusion. In Figure 5, SEM scans of the filaments revealed a smooth surface with no flaws, suggesting a complete optimised extrusion process. The layered structure of the 3D printed fruit chews was also investigated by SEM and presented consistent layer thickness varying from 99.3 to 125 μm.

### 3.4. Mechanical Studies

The mechanical properties of the extruded filaments play a major role in the printability of the filament formulations. Klucel ELF^TM^ is a well-known printable pharmaceutical grade polymer [25,35]. However, the addition of surfactants may result in excess polymer plasticisation which can affect the mechanical properties of the filament and eventually its printability [36]. Therefore, various ratios of placebo HPC-ELF and GLC blends were extruded and evaluated for the maximum tensile strength (MTS). As shown in Figure 6 the analysis revealed that the HPC-ELF/GLC at 95:5 (wt/wt%) ratio showed the highest rigidity and MTS while at 80:20 the performance was poor). According to our previous study, filaments with MTS of over 17 Mpa are considered printable while for filaments with lower values the printability remains a challenge for Bowden based FDM technology. On the other hand, for the drug loaded and placebo filaments (90:10 wt/wt%), the MTS were 18.9 MPa and 19.9MPa, respectively, rendering them suitable for printing. Such behaviour may be explained due to the presence of the API, flavour and sweetener, which improved the mechanical properties of the filament while maintaining the chewability of the 3D printed fruit chews.

### 3.5. Taste Masking and Sensory Evaluation

For the design of paediatric dosage forms the taste masking efficacy of the fruit chew printed structures is critical. The selected excipients, mainly GLP and HPC-ELF, were expected to facilitate excellent taste masking by eliminating the bitterness of DPH. The addition of sweeteners and strawberry flavour was anticipated to further enhance the palatability of the 3D structures and medication adherence for children. The colouring was added for aesthetic purposes to imitate confectionaries that are most preferable from paediatric populations. However, the taste-masking evaluation was performed in printed designs in the absence of any additives (sucralose and strawberry flavour). As shown in Figure 7, the printed designs presented excellent masking of the extremely bitter DPH, which scored 5/5 on the taste scale and where none of the subjects reported any unpleasant taste.

As mentioned before this was due to the extrusion processing which facilitated drug-polymer interactions through H-bonding interactions.

The next part of the study involved the sensory evaluation and the 3D fruit chews were examined for their sweetness intensity and strawberry aroma [32]. It is very important to identify the differentiation in sweetening intensity when the dosage is more complex, for example after the addition of flavours. Several studies have demonstrated the possible interactions between the sweeteners and food addictive which eventually alter the sweetener potency [37,38,39,40]. An important consideration in such studies is to identify the optimal ratio of sweetener/flavour including their total concertation in the dosage form. Herein, the optimised sucralose/strawberry ratio was 0.9:1.1 wt/wt% while several other ratios were tested (data not presented).

As shown in Table 2 the bulk sucralose presented a high sweet intensity with most of the subjects scoring “extremely sweet” intensity and a very strong aftertaste. For the 3D printed fruit chews the subjects observed a reduced intensity and most of them reported “very sweet” while a few noted “moderate sweet”. The perceived aftertaste was also recorded and subjects reported a reduced sweet aftertaste in comparison to the bulk sucralose.

Similarly, for the strawberry aroma evaluation of the bulk powder, the subjects scored a strong strawberry and fruit intensity with several reporting a sweet and aftertaste intensity for the particular flavour grade. Again, a range of various flavour grades (e.g., organic strawberry juice, strawberry juice powder) were tasted from different providers (data not shown) and the particular one was identified with the best “fruity” intensity. Interestingly, the evaluation of the 3D fruit chews revealed a strong sweet, strawberry, fruity intensity while most of the subjects also scored strong after taste. The results are attributed to the synergistic interactions between the sweetener and the strawberry flavour where the total intensity of the mixture is greater than that of the individual components. In addition, the strawberry and fruity perception of the printed fruit chews were not affected and remained strong despite the presence of other excipients in the final dosage form. Overall, the optimised sweetener/flavour ratio gave an excellent taste perception with enhanced palatability by imitating fruit confectionaries. It is worth mentioning that the combination of sweeteners/flavours of different grades and various ratios is unlimited and can result in completely different results.

### 3.6. Drug Release Profiles

The design of the extruded filaments was based on the selection of two hydrophilic excipients HPC-ELF and GLC, respectively. The former is a low molecular HPC grade while the latter is a hydrophilic non-ionic surfactant where both have found applications for immediate release dosage forms. DPH is a hydrophilic molecule and according to USP Pharmacopeia not less than 80% should be released within 45 min. As shown in Figure 8 the DPH release rates were rapid for all fruit-chew designs with a minimum of 85% (palm tree) been released within the first 30 min. As expected, a slight variation on the release rates of the different designs was observed with “smurfs” showing the faster release followed by decreasing order from the banana > cherry > palm tree > normal tablet. These variations are attributed to the different surface areas of the designs which results in varying dissolving rates of the hydrophilic carriers. The phenomenon has been observed previously by Goyanes et al. where the drug release rates are affected by the printed design [41]. In our case, the DPH release rates were aligned by adjusting the infill density at 30% while the rapid hydration of the carriers contributed to the fast dissolution rates.

Statistical analysis (Kruskal–Wallis nonparametric test followed by the Dunn post-hoc multiple comparison test) of the obtained release profiles showed no significant differences for all the printed fruit-chews. For the conventional tablet design, a slower initial DPH release rate was observed, due to the smaller total surface, but reached 83% within the first 30 min.

## 4. Conclusions

In the current study, personalised paediatric 3D printed fruit-chew designs were fabricated with effective taste masking and enhanced palatability. DPH was used as a model drug and co-processed with the hydrophilic HPC-ELF and GLC for the manufacturing of 3D printed filaments using HME processing. The formulation comprised of sweeteners and strawberry flavour was used to imitate confectionaries at predefined ratios of 0.9:1.1 respectively. The taste evaluation made by trained panellists of the extruded filaments showed excellent taste masking of the bitter DPH due to the formation of a DPH glass solution. In addition, the sensory results demonstrated sweet taste with enhanced strawberry aroma and fruity flavour taste for the fruit-chews with enhanced organoleptic properties. The synergistic effect of the sweetener and strawberry flavour also showed a very good aftertaste. The appropriate selection of the printable drug carriers and the adjustment of the infill density resulted in rapid DPH release with the first 30 min. In conclusion, this feasibility study demonstrated the successful development of personalised paediatric fruit-chews with enhanced palatability organoleptic properties that could help medication adherence in the future.

## Figures and Tables

**Figure 1 pharmaceutics-13-01301-f001:**
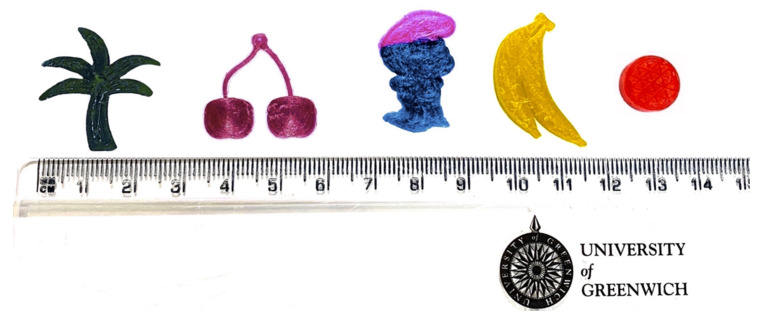
3D printed fruit chews. from left; palm, cherry, Smurf, banana and tablet.

**Figure 2 pharmaceutics-13-01301-f002:**
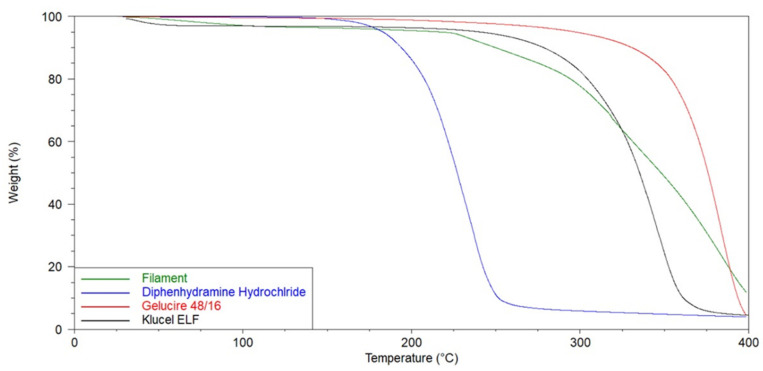
TGA thermograms of Klucel ELF^TM^, Gelucire 48/16^TM^, diphenhydramine hydrochloride and drug loaded filament.

**Figure 3 pharmaceutics-13-01301-f003:**
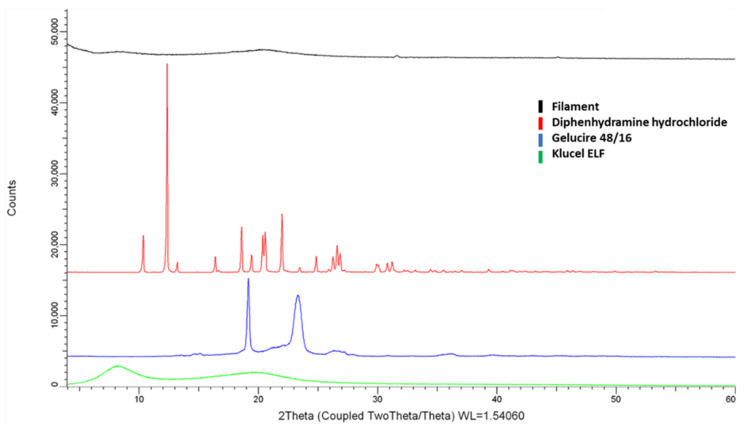
XRD patterns of Klucel ELF^TM^, Gelucire 48/16^TM^, DPH and drug loaded filament.

**Figure 4 pharmaceutics-13-01301-f004:**
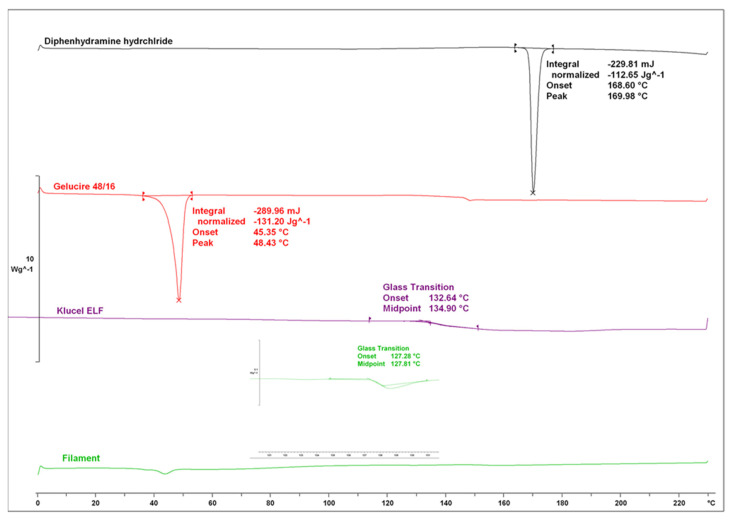
DSC thermograms of Klucel ELF^TM^, Gelucire 48/16^TM^, DPH and drug loaded filament.

**Figure 5 pharmaceutics-13-01301-f005:**
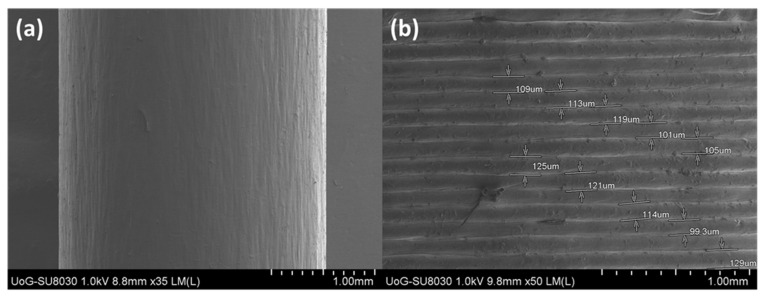
SEM images of (**a**) drug loaded 3D printing filament and (**b**) 3D printed fruit chew.

**Figure 6 pharmaceutics-13-01301-f006:**
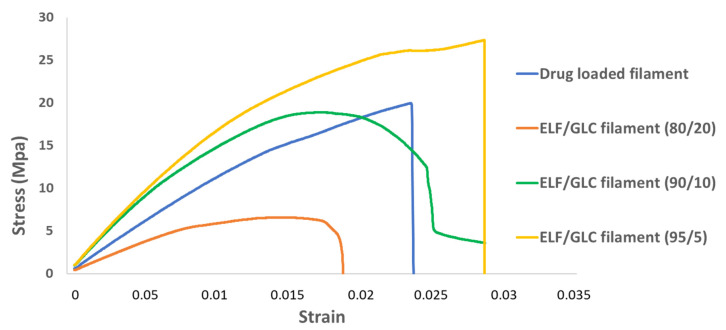
Stress vs. strain (Maximum Tensile Strength) graph of placebo drug loaded filaments.

**Figure 7 pharmaceutics-13-01301-f007:**
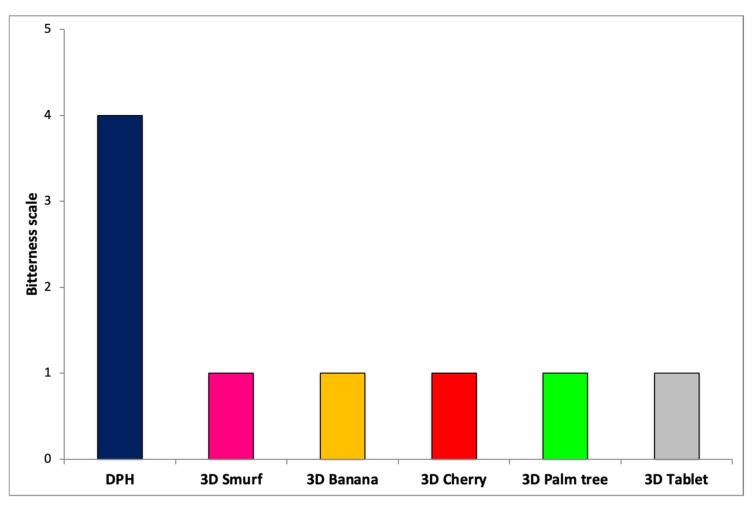
Taste masking evaluation of DPH and 3D printed fruit chews.

**Figure 8 pharmaceutics-13-01301-f008:**
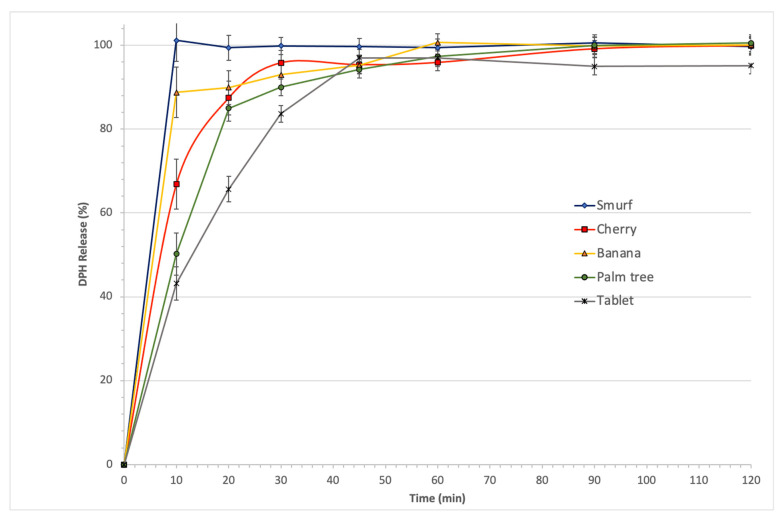
DPH release from 3D printed fruit chews in pH 6.5 for 2 h.

**Table 1 pharmaceutics-13-01301-t001:** Weight variation of the 3D printed fruit chew deigns.

Design	Average Print Weight (mg)
Palm	495 ± 23.5
Cherry	503 ± 18.6
Smurf	492 ± 29.1
Banana	505 ± 11.5
Tablet	501 ± 12.5

**Table 2 pharmaceutics-13-01301-t002:** Number of subjects reporting sweetness and strawberry aroma of bulk substances and 3D printed fruit chews.

**Sweetness**					
**Materials**	**No Sweet**	**Moderate Sweet**	**Very Sweet**	**Extremely Sweet**	**Aftertaste**
Sucralose			1–2	8–10	8–10
Fruit chew		1–2	8–10		6–8
**Strawberry Aroma**					
**Materials**	**Sweet**	**Strawberry**	**Sour**	**Fruity**	**Aftertaste**
STR	6–8	9–10		8–10	5–7
Fruit chew	8–10	9–10		8–10	8–10

## Data Availability

Not applicable.

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
