# Peer review of "Personalised Tasted Masked Chewable 3D Printed Fruit-Chews for Paediatric Patients"

_pharmaceutics, 2021, doi:10.3390/pharmaceutics13081301_

Round 1
Reviewer 1 Report
The manuscript that was suugested for review deals with the formulation of a dosage form intended for pediathric population. This is a very challenging task considering the specific target group of patients, their preferences and inclinations.
In this sense, the work is highly topical and the results obtained could contribute to the development of a formulation at high acceptance rate.
I have the following recommendations and questions to the authors:
1) The brand names Klucel ELF and Gelucire 48/16 should contain the symbol "TM"or registered mark.
2) Unnecessary capitalizing is noticed throughout the whole manuscript (line 19, line 93 and more - Sucralose; line 94 - Yellow; line 159 - Thermoplastic; and many many more. These should be corrected in the whole manuscript.
3) Abbreviation DHP, line 22 must be corrected.
4) Line 22 - 3d must be capitalzed.
5) Line 45-46 - the sentence should be revised.
6) The authors need to explain how the developed formulation refers to personalised dosage forms.
7) Line 64 - the word "tables" must be corrected.
8) Line 75 - I do not find it correct to use the brand name aspirin instead of the INN.
9) Line 77 - technologies must be in singular.
10) Line 79 - couple should be changed to coupled
11) Abbreviation for hot melt extrusion was used without being first introduced.
12) Line 96 - "was" should be replaced with "were".
13) Multiple misuse of the sign "period"was notised - line 100 - wt/wt%.; line 144 - The. D8, etc.
14) Line 142 - XPRD should be replaced with XRPD
15) The whole chapter 2.7 needs revision, the sentences are incomplete and cannot be understood.
16) In chapter 2.9 Dissolution studies line 166 filter pore size seems not correct. The authors should explain why they used 100 rpm and 6.5 buffer as acceptor media. Acording to USP Diphenhydramine capsules are evaluated in purified water using Apparatus 2 at 50 rpm.
17) Line 209 HLB abbreviation was not introduced.
18) Line 234 - douple typing of in; is the dose of the drug 2.5 or 12.5 mg.
19) Line 237 - must be due TO
20) Line 239 - the authors claim that the method was accurate and repeatable but no data was provided.
21) Line 241 - the authors claim "weight and content uniformity variation of 2-10% was detected"but the statement is not supported by data.
22) Line 279 - Diphenhydramin is not typed correctly
23) Line 318 - orange flavor is mentioned for the first time, strawberry flavour is mentioned before. Which is correct? The same in table 1, line 326 and lines 332, 339, 340.
24) Figure 2 - pulm tree is typed instead of palm
25) Line 360 - the authors claim that according to USP not less than 75% DPH should be released within 45 min but according to my inquiry, not less than 80% should be dissolved in 30 min. This should be clarified. http://www.pharmacopeia.cn/v29240/usp29nf24s0_m27140.html
26) Line 362 - minimum 85% been released within the first 20 minutes. In the abstract is said that 85 % are released in the first 30 min. Needs clarification.
27) Line 370 - the sentence is without ending.
28) Line 371 - the authors claim that no significant difference was shown in the release profiles. As we can see, there are substantial differences. How was statistical significance evaluated and what level was considered statistically significant?
29) references No 36 and No39 need to be presented in the same format as the others, without capitalization.
Author Response
The responses are attached in the Word file.

Reviewer 2 Report
Present manuscript deals with development of chewable 3D printed shaped dosage forms for paediatric patients. State of the art technology of 3D printing was used but the design of printed forms was unique. The manuscript quality should be improved. Some suggestions of improvement are given below.
- Page 3, line 123: dosage form weight should be given in a form of target mass +/- standard deviation
- Page 5, second paragraph (lines 197-202) should be moved to Introduction, and the following two paragraph (lines 203-223) should be move to appropriate site in Materials
- Page 6, line 239-241: Authors should add exact data on weight and content of individual printed form, their weigh and content variation – I suggest to add this data in a form of Table!
- Page 5 lines 242-252: Data on chemical purity of the raw materials (excipients and API) heated to max. working temperature of extrusion and printing should be added to show no degradation is occurring during the both processes.
- Figure 8: Authors should discuss and explain why drug release is decreasing after 45 minutes in case of “Tablet” sample.
- Page 15: correct writing of references 36 and 39 – why uppercase letters were used!
- The authors should add discussion for which age population theirs dosage forms are proposed in light of their size and weight.
Author Response

(The authors gave the same response as above.)

Round 2
Reviewer 1 Report
I have reviewed the revised manuscript and I am completely satisfied with the result. I have no further critical remarks.